# Night Vision and Carotenoids (NVC): A Randomized Placebo Controlled Clinical Trial on Effects of Carotenoid Supplementation on Night Vision in Older Adults

**DOI:** 10.3390/nu13093191

**Published:** 2021-09-14

**Authors:** Stuart Richer, Steven Novil, Taylor Gullett, Avni Dervishi, Sherwin Nassiri, Co Duong, Robert Davis, Pinakin Gunvant Davey

**Affiliations:** 1Captain James A Lovell Fed Health Care Facility, North Chicago, IL 60064, USA; livonlivon@yahoo.com; 2Chicago Medical School, Rosalind Franklin University of Medicine and Science, North Chicago, IL 60064, USA; taylor.gullett@my.rfums.org (T.G.); avni.dervishi@my.rfums.org (A.D.); sherwin.nassiri@my.rfums.org (S.N.); co.duong@my.rfums.org (C.D.); 3Davis Eye Care Associates, Oak Lawn, IL 60453, USA; eyemanage@aol.com; 4College of Optometry, Western University of Health Sciences, Pomona, CA 91766, USA

**Keywords:** automobile crash risk, cholecystectomy, glare disability, luminance preference, lutein, macular pigment optical density, night vision, zeaxanthin

## Abstract

Twilight and low luminance levels are visually challenging environments for the elderly, especially when driving at night. Carotenoid rich diets are known to increase macular pigment optical density (MPOD), which in turn leads to an improvement in visual function. It is not known whether augmenting MPOD can lead to a decrease in vision related night driving difficulties. Additionally, it is unknown if carotenoid supplementation provides additional measurable benefits to one’s useful field of view (UFOV) along with a decreased composite crash risk score. The aim of the study was to evaluate changes in night vision function and UFOV in individuals that took carotenoid vitamin supplements for a six-month period compared to a placebo group. Methods: A prospective, randomized, double-blind, six-month trial of a 14 mg zeaxanthin/7 mg lutein-based supplement was carried out. Participants were randomized into active or placebo group (approx 2:1). Results: *n* = 33 participants (26 males/7 females) participated with 93% capsule intake compliance in the supplemented group (*n* = 24) and placebo group (*n* = 9). MPOD (mean/standard error SE) in the active group increased in the Right eye from 0.35 density units (du)/0.04 SE to 0.41 du/0.05 SE; *p* < 0.001 and in the Left eye from 0.35 du/0.05 SE to 0.37 du, *p* > 0.05). The supplemented group showed significant improvements in contrast sensitivity with glare in both eyes with improvements in LogMAR scores of 0.147 and 0.149, respectively (*p* = 0.02 and 0.01, respectively), monocularly tested glare recovery time improved 2.76 and 2.54 s, respectively, (*p* = 0.008 and *p* = 0.02), and we also noted a decreased preferred luminance required to complete visual tasks (*p* = 0.02 and 0.03). Improvements in UFOV scores of divided attention (*p* < 0.001) and improved composite crash risk score (*p* = 0.004) were seen in the supplemented group. The placebo group remained unchanged. Conclusions: The NVC demonstrates that augmenting MPOD in individuals with difficulty in night vision showed measurable benefits in numerous visual functions that are important for night vision driving in this small sample RCT. Additionally, we observed an improvement in UFOV divided attention test scores and decreased composite risk scores.

## 1. Introduction

The mean population age in developing countries, as well as developed countries like the United States of America, is increasing. The risk of automobile injury or fatality (driver, passenger, or pedestrian) associated with motor vehicle accidents has been determined to increase with age [1]. This is due to a complex interplay of age-related declines observed in vision, motor, and cognitive functioning [1]. Furthermore, basic visual function is the foundation for higher order neural processing, whether it is visual–spatial or cognitive.

Aging and increasingly immobile populations create unmet clinical, societal, and economic burdens. Recent reports reflect on the cost of poor vision on driving [2]. A recent assessment of visual impairment among European drivers examined 2422 drivers from five European countries [3]. Visual acuity, visual field, contrast sensitivity, glare sensitivity, and useful field of view (UFOV) were tested. Broader visual functions not included in the current licensing standards were found to be more impaired among drivers (compared to those functions legally required) [3]. Elderly drivers are particularly vulnerable to sensory visual impairment when driving at night, as they suffer declines in contrast sensitivity (CS), glare disability (GD) and glare recovery (GR) [3,4,5].

In the United States of America, the lifetime economic costs of ageing-related accidents associated with driving is $150 billion annually as per the National Highway and Traffic Safety Administration 2020 report [6]. Additionally, older Americans are overrepresented in traffic fatalities. Their fatality rate is 17 times as high as the 25–65 year age group [6].

The NEI AREDS2 study [7] substantiated the safety and usefulness of prescribing carotenoids (10 mg Lutein/2 mg zeaxanthin) in patients at a high risk of age-related macular degeneration or AMD [7]. Significantly, the average American and veteran patient are typically low dose consumers of these carotenoids (1 to 2 mg/day). Most of the world is no different, except China where dietary intake of lutein is higher [8]. Prior work by our labs have demonstrated the efficacy of carotenoid supplementation and macular re-pigmentation for improving the self-described driving ability of patients with AMD [9]. Our data was affirmed by other investigators internationally, and 23% of car accidents are associated with drivers’ reduced visual performance [10].

The current study is a prospective randomized clinical trial (RCT) with participants who are not professional drivers or pilots, have difficulty with night vision, and are free of overt disease but self-report diminished night vision. The Night Vision Carotenoid Study (NVC) focuses on the assessment of increasing MPOD by nutritional supplementation and associated improvements in vision (i.e., low luminance visual function, contrast sensitivity and disabling glare/diminished glare recovery (i.e., due to headlights)). Prior publications have evaluated this area [11,12,13,14], however, we additionally looked at how UFOV and visual attention differ in individuals with high vs. low MPOD [15]. The causes of motor vehicle accidents can be explained by a slowing of brain responses, impaired judgment, and the deterioration of the vision of drivers [16]. The xanthophylls lutein and zeaxanthin comprise greater than 70% of the xanthophylls in the brain, with MPOD potentially becoming a surrogate measure of brain carotenoids and brain function [17,18]. Significantly, lower MPOD is associated with poorer performance on the mini-mental state examination and the Montreal cognitive assessment [19]. Individuals with lower MPOD have poorer prospective memory, take a longer time to complete a trail-making task, and have slower and more variable reaction times in choice reaction tasks [19]. Additionally, intake of the carotenoid zeaxanthin is known to improve visual memory [20]. Therefore, we incorporated three distinct measures: UFOV, visual processing speed, and “crash risk”, a composite measure in the present study.

The objectives of the study were to determine whether a ‘once per day carotenoid nutritional supplement’ that aids in augmenting MPOD decreases night-time driving difficulty in individuals free of overt ocular pathology. Given past literature on the benefits of carotenoids and visual function and its benefits in various disease states [7,9,10,11,12,13,14,15,16,19,20,21,22,23,24,25,26,27], we hypothesized that the intake of carotenoid vitamin supplements would show measurably improved visual performance compared to a placebo group.

## 2. Materials and Methods

The NVC study is a prospective six-month placebo controlled randomized clinical trial (RCT) evaluating the intake of a nutritional supplement taken once per day with food containing 14 mg zeaxanthin/7 mg lutein and additional non-carotenoid vitamins/minerals (ScreenShieldPro^®^, EyePromise, Chesterfield, MO, USA; see Appendix A) on macular pigment and night vision parameters. The study population were US veterans at the Captain James A Lovell Federal Health Care Center (FHCC) between 1 August 2017 and 1 August 2019, with participants sequentially recruited and enrolled. The 2:1 active:placebo randomization was selected as it is difficult to find veterans at our medical center who are not using carotenoid supplements, and we wanted to obtain a larger sample for the supplement group. Additionally, carotenoids cannot be synthesized by our bodies and need to be acquired nutritionally, so it is unlikely that the placebo group would show marked variation.

The MPOD at one-degree surrounding fovea was the independent variable determined at a visit schedule of four time points (baseline, 6, 12 and 24 weeks). *n* = 33 veterans, mean age (±SD) of 62.7 (10.8) years without age related macular degeneration or clinically significant cataract. Given that the visual function and performance can be influenced due to monocular differences, the right and left eyes of individuals were analyzed separately and not grouped unless the test needed to be performed binocularly. To be included in the study, participants had to have a LogMAR acuity of 0.01 or better in both eyes, and additionally meet at least two of the four criteria: (1) MPOD < 0.20 du in one or both eyes, (2) low luminance 32-item questionnaire failure [28,29], (3) retinal OPTOS foveal/parafoveal/RPE autofluorescence defect in either retina, or (4) Adapt Dx score of <6.5 min [30]. Individuals were excluded if they met any of the following criteria: (1) Had taken any supplementary carotenoids within 10 weeks of baseline visit; (2) had significant media opacity, cataract or congenital/acquired retinal disease other than minimal atrophic changes, cataract, or diabetic retinopathy; (3) had any recent ophthalmologic surgery or treatment; (4) medically/psychiatrically unable to take part in a six-month study; or (5) were prescribed (self or professionally) an over the counter (OTC) product purported to improve their night vision (i.e., containing carotenoids, polyphenols like bilberry, etc.). The inclusion of the participants was determined by the principal investigator (SR).

Active and Placebo Capsules: The active was a 21 mg dietary carotenoid supplement containing 14 mg zeaxanthin/7 mg lutein and additional antioxidants and minerals specified in Appendix A. The placebo capsule was a similar appearing inert capsule containing high-oleic safflower oil and did not contain any other antioxidant or minerals. The active and placebo groups were prescribed by pharmacy prescription, following inclusion, exclusion, and initial baseline data acquisition. Following written informed consent, the FHCC research pharmacist dispensed a single bottle of gel caps to each participant.

### 2.1. Data Acquisition

Study participants were asked detailed questions about their smoking history (pack/years), history of cholecystectomy, any over the counter medications they had taken, including dietary fat blockers or acid blockers. All participants had body fat measured using bioelectric impedance (15 to 40% + range).

### 2.2. Instrumentation

Refraction: Best corrected visual acuity was determined with a Nidek RT-5100 Refracting system paired with an MS&S Smart System Digital Projection System [31,32]. The refractive correction obtained was used at baseline, and the best visual acuity was determined for each eye and used on all four visits. All research tests were performed monocularly per the manufacturer’s recommendations, except when indicated by the manufacturer.

Low Contrast Snellen Visual Acuity Letters: An Office Snellen Visual Acuity at three different contrast thresholds (M&S technologies, Chicago, IL, USA) (6%, 12%, or 25%) two lines above the maximum best refracted visual acuity was measured using M & S computerized charts. The participants sequentially viewed lines of 6%, 12.5%, and 25% contrast Snellen letters on a computerized projection system through the best refraction with each eye and identified which threshold line they could read.

Macular pigment optical density (MPOD): MOPD was measured at the central degree of the fovea. MPOD is a measurement of the attenuation of blue light by macular pigment and is linearly related to the amount of carotenoids one consumes [21]. MPOD was determined by the psychophysical method of heterochromatic flicker photometry [33,34]. The QuantifEye^®^ is the clinical gold standard with excellent repeatability [11,21,33,34] and good correlation with other objective techniques [35,36,37] that has been used in numerous clinical studies [9,21,22,26,27,29,33,34,35,36,37,38]. The test is only performed on one eye at a time, and the eye not being tested is patched. A one-degree target that is composed of two alternating lights of different blue and green wavelengths is presented. The blue light is absorbed by the macular pigment, whereas the green light is outside the absorption spectrum of macular pigment. The participant perceives a flicker and is asked to click the button when they see the target flicker. A computer program calculates a curve wherein the MPOD is based upon the amount of blue/green ratio in the stimulus with an additional compensation for the natural age-related yellowing of the lens based upon an age normative algorithm. The MPOD is recorded in density units (du) for each eye.

Vimetrics Central Vision Analyzer (CVA): This test is a measure of contrast and glare disability. The test is an interactive computer device that measures threshold visual acuity by presenting a Landolt-C with the gap in the C randomly tumbled to one of four cardinal positions to which the patient responds with a four directional response pad [29]. The threshold is approached in a logarithmic, staircase fashion to define the test participant’s acuity threshold with a resolution similar to the letter by letter scoring of CSF/Glare chart testing but with the ability to rapidly test under a number of conditions of luminance and contrast those with the activities of daily living [39].

LuxIQ preferred luminance: The LuxIQ is a calibrated exam tool providing lighting assessment recommendations. The preferred luminance was established on a desktop calibrated device consisting of four arrays of high-brightness LEDs providing uniform illuminance in a 14 × 20 cm viewing area that systematically evaluates lighting preferences over a broad range of light levels (0–5000 lux) and color temperatures (2700–6500° K) [40]. The LuxIQ™ was placed over a visual acuity chart using the individuals’ near working distance based on the near refraction. The color temperature was fixed at a maximum impairment (6500° K), and the participant asked to find the preferred luminance three times. This desktop test quantifies the photophobia of bright daylight driving conditions and extreme blue headlight glare. The average preferred luminance was recorded for each visit.

MDD-2 photo- stress glare recovery: Photo-stress recovery time is the time taken for visual acuity to return to normal levels after the retina has been bleached by a bright light source. The MDD2 device is a handheld monocular flash instrument that is used to differentiate normal from abnormal macula, detecting worsening macular disease [41]. Participants wearing refractive correction look through a 12 mm aperture into a flash tube held close to the eye. Within the tube, they see random alphanumeric stimuli at an apparent size of 0.41 radian angular subtense. Following a white flash stimulus, the “time to recovery” in seconds is recorded with built-in timing circuitry. The measurements obtained by MDD2 are known to be repeatable, providing a potential clinical biomarker of ocular health [42]. Due to the visual disabling nature of this test, which simulates headlight glare, photo stress recovery MDD-2 testing was accomplished as the very last test in each research session.

UFOV (useful field of view): In human vision, the useful field of view (UFOV) is the visual area over which information can be extracted at a brief glance without eye or head movements. The UFOV computer simulation was accomplished using BrainHQ^®^ UFOV software on a 15-inch monitor with near correction [43]. Brain HQ is a test of functional vision and visual attention; these can be predictive of the ability to perform many everyday activities, such as driving a vehicle. The test is administered in about 15 min and is recommended for use as a screening measure in conjunction with a clinical examination of cognitive functioning or fitness (i.e., manual dexterity) to drive. UFOV consists of three subtests that assess the speed of visual processing under increasingly complex task demands. Using both eyes, the examinee must detect, identify, and localize briefly presented targets. In the first subtest, Processing Speed, the examinee identifies a target presented in a centrally located fixation box that is presented for varying lengths of time. In the second subtest, Divided Attention, the examinee identifies a target, but must also localize a simultaneously presented target displayed in the periphery of the visual field. The third subtest, Selective Attention, is identical to the second, except that the target displayed in the periphery is embedded in distractors, making the participant’s task much more difficult. An interpretive report provides scores for each part of the UFOV and assigns the individual to one of five levels, or categories, of risk. According to the manufacturer, UFOV may be used to aid in making professional judgments about an individuals’ fitness to drive.

Dark Adaptometry: Dark Adaptometry was performed using Maculogix AdaptDx, which calculates a simplified, objective measure of dark adaptation speed (in minutes) called rod intercept (RI). It has been traditionally used to diagnose age related macular degeneration [30]. The dark adaptometry measurements were performed with pupils dilated (≥6 mm). Pupillary dilation was achieved using 1% tropicamide, and 2.5% phenylephrine HCl solution. Dilated pupils were exposed to varying intensities of blue–green light at mixed intervals to calculate the RI, which is the amount of time it takes for the eyes to adapt to darkness. Any RI < 6.5 min is normal dark adaptation function whereas ≥6.5 min indicates impaired dark adaptation.

## 3. Statistical Analysis

Power calculation: A human, double blind, randomized, placebo-controlled intervention and “intention to treat” study was based upon a power calculation with approximately 80% power and a two-sided alpha error of 5%. The primary measure of successful intervention was to obtain a 10% increase in the MPOD measurement compared to the control group. The current study with a treatment group of *n* = 24 was sufficiently powered to obtain statistically significant changes in MPOD.

All continuous variables are expressed as mean ± standard deviation; categorical variables are expressed as absolute numbers and percentages. Differences in continuous variables between control and treatment groups in baseline characteristics are analyzed by independent sample *t*-test and categorical variables are analyzed by Fisher’s exact test. Participants were evaluated at baseline, six weeks, three months, and six months. Two-sample *t*-tests were used to examine the differences between the baseline and measurements at the final visit.

Randomization was achieved with full blinding of all FHCC NVC study personnel and participants. A grant sponsor center employee whose identity was unknown to all study personnel and participants electronically used Microsoft Excel’s RAND formula features following a two-step process to randomize participants as to the treatment and placebo group. Once the randomization was completed, an approximate 2:1 (active:placebo) ratio was accomplished. All investigators were blinded as to the treatment arm. The grant sponsor maintained the identity of the study’s four-digit numeric code that was released to the statistician following completion of the study. NVC was registered as clinical trial IRB 1052607-1 at Hines VAMC and as NCT04741763 at www.clinicaltrials.gov.

## 4. Results

Table 1 provides baseline parameters of the study participants. The placebo group had nine participants (seven males and two females). The treatment group had twenty-four participants (19 males and five females). The supplement group significantly (*p* < 0.05) differed from the placebo group in age. The waist to height ratio, BMI and body fat percentage were greater in the supplement group when compared to the placebo group and showed a tendency towards statistical significance (*p* = 0.05, *p* = 0.06 and *p* = 0.06, respectively). There were also a higher number of participants in the supplement group with a surgically removed gallbladder (cholecystectomy)—eight participants vs. only a single individual in the placebo group. This was, however, not statistically significant (*p* = 0.27 Chi Square). The various baseline parameters of the Vimetrics mesopic tests and the low contrast tests and the amount of cataract as judged by LOCS grading scale were not significantly different between the placebo and the treatment group (*p* > 0.05 for all parameters).

### 4.1. Macular Pigment Optical Density

Ninety three percent capsule intake compliance was recorded by the investigators. It must be noted that the mean baseline MPOD of the placebo group was greater than the baseline MPOD of the treatment group. Both the right eyes and left eyes of the supplement group showed an increase in MPOD, although interestingly the increase in MPOD was asymmetric between the eyes. The right eyes of the supplement group showed an increase in MPOD (mean/SE) 0.35 du/0.04 to 0.41 du/0.05, whereas left eyes’ mean MPOD changed from 0.35 du/0.05 to 0.37 du. There was a 17% increase in the MPOD of the right eye and only a 6% increase in the MPOD the left eye. The change in the mean MPOD in the right eye was statistically significant whereas the MPOD increase in left eye did not reach statistical significance (*p* < 0.001 and *p* > 0.05, respectively). There was no change in MPOD in the placebo group (Figure 1A,B).

### 4.2. Glare & Contrast Improvement

A Vimetrics Central Vision Analyzer (CVA) was used to measure three levels of contrast challenge (100%, 64%, and 43%) and glare disability challenge (100%, 10%, and 8%). Comparing the baseline data to the results at six months using a two tailed *t*-test accounting for unequal variance, we find that the supplement group showed a tendency to improve in the contrast sensitivity function at 64% contrast levels under mesopic viewing conditions (*p* = 0.06) and glare disability challenge in performing contrast sensitivity under a high glare situation, *p* = 0.02; 0.01, respectively for right and left eye. The mean LogMAR values for baseline contrast sensitivity with glare 100 was 0.20 and 0.19 for right and left eyes that improved to 0.05 and 0.03, respectively, with six months of carotenoid vitamin supplementation.

### 4.3. Glare Recovery Improvement

Figure 2 presents photo stress glare recovery data in seconds using the MDD2 device. The values obtained at the six-month time point were compared to the baseline. The supplement group showed glare recovery times twice as fast after carotenoid vitamin supplementation at six months in both eyes compared to baseline data (*t*-test *p* = 0.008 and *p* = 0.02 right eye and left eye, respectively). Although there was an improvement in glare recovery time, the placebo group did not change significantly compared to baseline in both eyes, indicating a possible learning curve in testing (*p* > 0.05, respectively).

### 4.4. Preferred Luminance in Lumens

Preferred luminance in lumens chosen, or the ideal subjective luminance is a monocular test, viewing a scene under a highly stressed 6500° blue kelvin color temperature light source, simulating bright daylight driving conditions. The participant made three adjustments of preferred luminance that were averaged. Figure 3 shows the right eye and the left eye results for the treatment and the placebo groups. The supplement group showed significant improvements and were able to see better with less lumens in both right and left eyes compared to the baseline data (*t*-test *p* = 0.02 and 0.03, respectively). Meanwhile, the placebo group did not show any significant changes in either eye compared to the baseline data (*t*-test *p* > 0.05, respectively).

### 4.5. Useful Field of (UFOV) Reaction Times

Figure 4A–C show the UFOV reaction times for central processing speed, reaction times for divided attention and selective attention, respectively. The central visual processing speed in the treatment group initially delayed but eventually equalized with the placebo group. Although there was quickening in the central visual processing speed from mean (SE) 20.9 msec (2.71) to 17.7 msec (0.69), it was not statistically significant (*t*-test *p* > 0.05). The divided attention reaction time task (Figure 4B) in the supplemented group improved from 150 msec (20.8) to 87 msec (11.2), significant at (*p* < 0.001). The selective attention time (Figure 4C) also quickened from 210 msec (24.6) to 158 msec (18.3), although not significantly (*p* < 0.10). The imputed composite crash risk score (1 to 3 scale) that takes into account all three visual processing scores showed a statistically significant improvement during the study duration from a mean (SE) of 1.96 (0.19) to 1.29 (0.11) in the supplemented group (*t*-test *p* = 0.004).

### 4.6. Low Contrast Snellen Visual Acuity Letters

We did not find low contrast exam room acuity to be sufficiently sensitive to detect changes in night vision over time with low contrast visual acuity not changing significantly in either group (6%, 12%, or 25% contrast optotypes threshold, *t*-test *p* > 0.05).

### 4.7. Retinal Fundus Auto-Fluorescence

In exploring other objective measures of night vision dysfunction, we hypothesized that retinal fundus auto-fluorescence (FAF) imaging might provide structural hidden information about the health and function of the central retina and paracentral retina. Accumulation of lipofuscin is associated with subtle retinal degeneration and often night vision difficulties [44]. Fundus autofluorescence defects in different retinal quadrants were examined with an OPTOS 200TX wide field camera and exported from the Optos V^2^ Vantage Pro Review software for further analysis.

An early treatment diabetic retinopathy study (ETDRS) grid overlay was applied to the macular region of the autofluorescence images obtained from the OPTOS camera with Adobe Photoshop CS3 software. According the ETDRS, the macula was delineated into nine macular sectors with three concentric circles of diameters 1 mm, 3 mm, and 6 mm to represent the fovea, parafovea, and perifovea, respectively (Figure 5A). Parafoveal and perifoveal rings were further divided into four quadrants to represent the superior, nasal, inferior, and temporal regions [45]. Standardization of the autofluorescence images involved: rotational alignment across all visits per eye for each participant, a contrast level of 100, and equalization-size scaling. Quantification of each macular sector was performed on Adobe Photoshop CS3 Software and recorded as the pixel density. Macular density measurements of each eye were plotted on a line graph over time for the following visits: baseline, six weeks, three months, and six months. All outliers were reviewed and re-quantified, whereas poor quality images were omitted from the quantification. Figure 5B represents the baseline R and L eye FAF values for both active and placebo groups. There were no differences in retinal FAF between the groups. Thus, we found this simple clinical test unhelpful for discerning differences in visual function over time.

## 5. Discussion

This study found that an improvement in MPOD with six-month carotenoid vitamin supplementation in individuals without overt retinal pathology but with difficulty in night vision and driving led to improvements in contrast sensitivity with high glare, glare recovery times, preferred luminance levels, useful field of view (UFOV) reaction times, and composite crash risk scores. The supplement group also showed a greater subjective improvement in vision as assessed by low luminance questionnaire compared to the placebo group. Driving is indeed a complex task involving physical, visual, and cognitive skills as well as manual dexterity. Although dexterity can be tested with driving simulators, it was considered beyond the scope of the NVC study, and the study concentrated on vision and reaction times. We had hypothesized that improvements in MPOD would lead to both improvements in vision and visual processing. The statistically significant improvements in various visual function and reaction time parameters tested supports our primary hypothesis.

Numerous studies have shown that there are various visual benefits from augmenting MPOD in healthy participants and in individuals with diseases like AMD, diabetes, and glaucoma—seen as enhanced contrast sensitivity, glare recovery, color discrimination, etc. [7,9,11,12,14,20,21,22,23,24,25,26,27,29,42,46,47,48]. The results observed in NVC buttresses the literature and demonstrates that improving MPOD with a carotenoid nutritional supplement decreases difficulties in night vision and driving. This is potentially important if patients have low MPOD to begin with and are not on carotenoid vitamin therapy for other reasons.

The relatively small sample size of the study was due to the difficulty in finding individuals that are not on OTC vitamin supplements. In studies that have a difficult-to-obtain sample, a 2:1 active versus placebo arm random distribution is common practice. Despite the small sample size there were obvious improvements observed in the treatment arm with statistically significant improvements in measured MPOD, visual and UFOV reaction times indicating that the sample size was indeed sufficient for the effect size observed.

We found that MPOD increased in both eyes of the supplement group, although the increase in MPOD score was asymmetrical, which is intriguing. In ocular healthy individuals, the MPOD is very similar between the eyes and with nutritional supplements we see a similar MPOD increase in both eyes [33,49]. With nutritional supplementation, an asymmetric increase in MPOD between the eyes was seen with a recent RCT that evaluated individuals at risk of AMD who had asymmetric retinal damage [22]. The participants of the present study did not have overt pathologies but did have some fundus autofluorescence changes, and individuals with minimal atrophic changes or diabetic retinopathy were included, thus they were not technically free of pathology. The asymmetric MPOD increase could be attributable to subtle differences in retinal health between the eyes. One could also postulate that aging causes changes in retinal physiology that leads to the physiological retinal uptake of carotenoids being asymmetric between the eyes. This could in part explain why most chronic age-related eye diseases like AMD, glaucoma, and diabetic retinopathy, whose pathophysiology includes significant oxidative damage, are asymmetric between the eyes of an individual. Future studies are needed to explore the possible reasons for asymmetric uptake of carotenoids and longitudinal studies are needed to investigate if a primary decrease in MPOD without clinical damage in the elderly leads to subsequent asymmetric age-related pathology.

Our previously published work suggests that carotenoids enhance cognition [20], which is attributed to adding UFOV testing. This is directly related to visual processing, on the road driving results, and a valid predictor of road accidents [1,50]. Notably, UFOV reaction times showed improvements in the supplement group over six months in divided attention, and imputed crash risk score. There was also a tendency towards improvement in selective attention, which invites further investigation.

The improvements in glare disability and contrast for the more challenging stimuli are consistent with the published literature and our previous publications and research on early age-related macular degeneration [26,27]. There is a new carotenoid test called the “Veggie Meter” that uses reflection spectroscopy to measure skin carotenoids. The Veggie Meter has shown an inverse relationship with skin levels of carotenoids, weight, and BMI [51]. The BMI of the supplement group of our study was greater than the BMI of the placebo group. Furthermore, the percent fat as measured using the bioelectric impedance was over 5% greater in the supplement group compared to the placebo group. The greater percent body fat is known to be a significant barrier in augmentation of MPOD as serum carotenoid is accumulated in the adipose tissue rather than being available to end organs like the eye and brain. Despite the poorer characteristics of BMI, the percent body fat of the supplement group compared to the placebo group showed a significant increase in measured MPOD for the supplement group. This is probably due to the fact that the dose of carotenoid in the treatment group was higher than other popular supplements like AREDS-2 supplements, which led to an increase in MPOD after six months.

The placebo group had one individual with a history of gallbladder removal (cholecystectomy), whereas there were eight such individuals in the treatment group. For carotenoids to be well absorbed into the circulation, they must be incorporated into mixed micelles, which is a mixture of bile salts and various lipid foods. Therefore, individuals that have undergone cholecystectomy may have difficulties absorbing carotenoids. In this study, we find that despite the great number of individuals in the treatment group with a history of cholecystectomy, six-month supplementation with carotenoid vitamin supplementation led to a measurable and significant increase in MPOD in their eyes.

The mesopic contrast visual acuity modules improved across multiple time periods (*p* < 0.07 for trend) except for high contrast stimuli. This is consistent with the excellent visual acuity of the study participants, and, as found in other carotenoid supplementation studies, it is difficult to improve beyond acuity with smaller letters than 20/20. Indeed, more subtle aspects of visual function, typically not evaluated in either the exam room or at driving licensing facilities, were seen in the supplemented group.

The MDD2 improvement in glare recovery is also consistent with the literature [11,12,14] and our previous publications of patients with early age-related macular degeneration improving their photo-stress glare recovery performance [26,27]. Wilson et al. recently noted that MPOD levels were significantly higher in individuals that reported no discomfort in or around the eyes than in those reporting mild discomfort in a non-parametric correlation analysis between NEI/Visual Function Questionnaire (VFQ-25) and surveys about glare and discomfort [52].

The improvements seen in glare disability, mesopic contrast, glare recovery, and preferred luminance with carotenoid supplementation supports and agrees with prior studies that have hypothesized that greater MPOD leads to a decreased scattering of light and a subsequent reduction in discomfort from glare [11,12,14,21].

In searching for a simple objective measure of night vision dysfunction, we also explored the hypothesis that retinal autofluorescence might be of assistance to eye practitioners with competing priorities beyond testing night vision. This test was included in NVC as a large number of optometrists (approximately 6000 practitioners) have such technology at their disposal. Unfortunately, it was not found discriminatory in placebo vs. participants supplemented with carotenoids. Similarly, we did not find The AdaptDx^®^, a proven sensitive and specific test for age-related macular degeneration retinal pigment epithelial dark adaptation, to be statistically discriminatory in evaluating healthy individuals with night vision difficulty. However, unlike patients with AMD that have increased rod-intercept (RI) times, the study participants were already at RI < 6.5 min and they would not be expected to improve much further.

NVC shows that enhancing retinal macular pigment optical density through nutraceutical supplementation reduces the blinding effects of glare and improves visual–cognitive driving performance skills. The NVC confirms the conclusions of previous investigators that increasing macula pigment can benefit fundamental first-order visual function abilities such as contrast sensitivity function, glare disability, glare recovery, luminance comfort, and visual as well as cognitive mediated reaction times, which are so critical to driving.

The study indeed has some limitations. The study is a single center trial with a relatively small sample size. The mean age of the placebo group was four years greater than the treatment group, which was statistically significantly. Similarly, the mean MPOD of the placebo group was greater than the treatment group. These differences between the means of the groups are likely due to the small sample size of the study and the randomization effects not being able to get a good balance. This should not have any bearing on the study results as the groups were compared longitudinally with themselves and not with each other. The sample size was indeed sufficient to show statistical significance and trends, but larger multi-center trials with larger sample sizes are needed. The parameters of percentage body fat, BMI and cholecystectomy count was not statistically significantly different between the placebo and the supplement group, but there was a notable difference that was almost significant. However, it should be noted that these undesirable factors possibly decrease the effect size observed in the supplement group and thus the results observed provide only conservative estimates of the benefits of the nutritional supplements.

## 6. Conclusions

In this study, we found that treatment with a carotenoid vitamin supplement caused an improvement in MPOD by 0.06 density units in individuals with initially low macular pigment and who were further disadvantaged by increased adiposity/cholecystectomy (reducing their carotenoid/fat soluble vitamin absorption). Furthermore, these individuals demonstrated improvements in multiple aspects of visual function, UFOV scores, and imputed driving crash risk composite scores when compared to a placebo group. The practical implication of this study is that a once daily capsule of a zeaxanthin/lutein-based nutritional supplement shows promise in augmenting night vision and improved driving performance in individuals that have difficulty driving.

## Figures and Tables

**Figure 1 nutrients-13-03191-f001:**
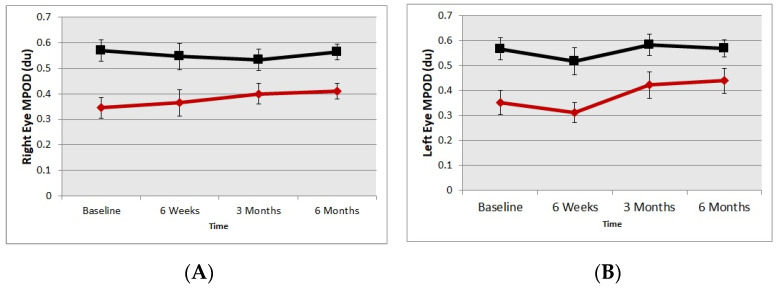
Macular pigment optical density change over time. **A** = right eye; **B** = left eye. The red diamond symbols represent Treatment group and black square symbols that represent the placebo group.

**Figure 2 nutrients-13-03191-f002:**
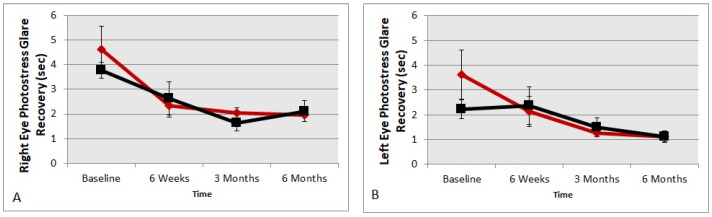
DD2 glare recovery improvement. Photo stress glare recovery data in seconds using the MDD2 device. **A** = right eye; **B** = left eye. The red diamond symbols represent the treatment group and black square symbols represent the placebo group.

**Figure 3 nutrients-13-03191-f003:**
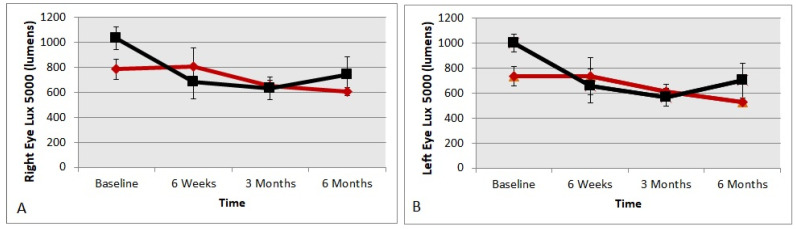
Preferred luminance in lumens chosen or ideal subjective luminance for right eyes and left eyes, over time, viewing a highly stressed 6500° blue Kelvin color temperature light source, simulating bright daylight driving conditions. **A** = right eye; **B** = left eye. The red diamond symbols represent the treatment group and black square symbols represent the placebo group.

**Figure 4 nutrients-13-03191-f004:**
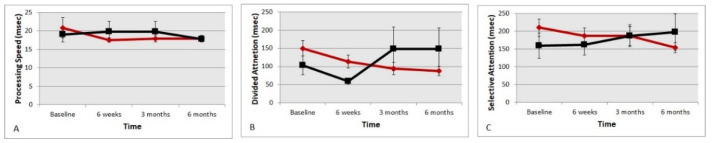
(**A**–**C**) Changes in reaction times seen in processing speed, divided attention and selective attention using the brain IQ useful field of view (UFOV). The red diamond symbols represent the treatment group and black square symbols represent the placebo group.

**Figure 5 nutrients-13-03191-f005:**
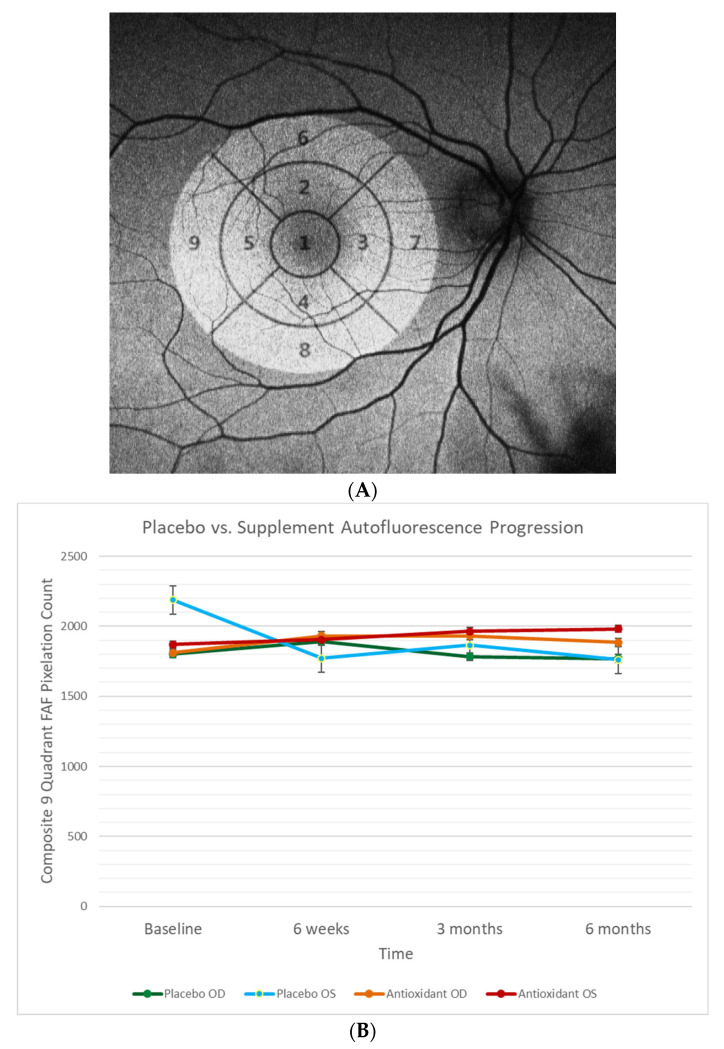
(**A**) Standard national eye institute ETDRS (early treatment of diabetic retinopathy) grid overlay on an OPTOS FAF image. (**B**) Nine quadrant composite pixilation count (placebo and active) over four time points, for OD (R eyes) and OS (L eyes). There was no significant difference between the active and placebo groups in either eye in terms of diminution of retinal auto fluorescence.

**Table 1 nutrients-13-03191-t001:** Baseline parameters of the study participants.

	Placebo Group	Treatment Group	*p*-Value
	Mean (Standard Deviation)		
Age	65.7 (8.0)	61.6 (11.63)	**0.027**
Body Mass Index	26.3 (3.42)	31.30 (6.54)	0.06
Percent Body Fat	28.2 (4.56)	33.5 (6.81)	0.06

All comparisons were done using independent *t*-tests with unequal variance *p*-value in bold are statistically significant *p* < 0.05.

## Data Availability

Data are contained within the article.

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
