# Peer review of "Night Vision and Carotenoids (NVC): A Randomized Placebo Controlled Clinical Trial on Effects of Carotenoid Supplementation on Night Vision in Older Adults"

_nutrients, 2021, doi:10.3390/nu13093191_

Round 1

Reviewer 1 Report

Thank you for giving me the opportunity to review this interesting work. Aging comes with its attendant challenges including a decline in vision. Consequently, providing a form of intervention through carotenoid supplementation is a promising way of addressing vision challenges that could impair safe driving. This work is thus warranted. That notwithstanding, kindly find below my comments for your perusal.

Title: The Night Vision and Carotenoids (NVC): A Randomized Placebo Controlled Clinical Trial. I would like to suggest that, the authors consider revising the title to “The effect of carotenoid supplementation on indices of improved night vision on older adults: A Randomized Placebo Controlled Clinical Trial (The Night Vision and Carotenoids (NVC) study)”

Abstract

Line 18: Please use “The aim of the study…”

Line 20: Please, replace “Double masked” for “double blind”. This is because throughout the manuscript, the authors have consistently used “blinding”.

Line 22: The author uses “subjects” yet throughout the manuscript, the author uses “participants”. I would like to suggest that, the authors consider using participants throughout for consistency sake.

Line 27 and 28: Kindly indicate what the magnitude of the improvement in the binocular contrast sensitivity with glare was?

Line 34: Kindly replace “divded” with “divided”

The conclusion must state by how much in magnitude those changes occurred.

Introduction

Line 42: The authors should kindly clarify this “Is the US a deveoloping country or a developed one?”

Line 64: This statement “The NEI AREDS2 study, May 2013” needs to be re-written. Do the authors mean it was published in May 2013 or it was released on that date?

Line 84-85: Please revise this statement “The causes of motor vehicle accidents are in part influenced by slowing of brain responses, impaired judgment the deterioration of vision of the drivers”.

Line 95: Kindly replace “NVC” with “the study”

Line 97: The authors should please replace “……past literature of benefits….” with “ …..past literature on benefits…”

Materials and Methods

Line 103-106: The sentence isn’t a complete sentence. The authors should revise it. Authors should revise it to “The NVC study…..evaluating the intake of …..on macular pigment and night vision parameters”.

The authors indicated that their participants were regular patronisers of dietary supplements. How did they ensure that, the effect seen was not attributed to the intake of the other supplements? They have stated that in the exclusion criteria but the emphasis was on “carotenoid” whereas in most supplements it’s “vitamin A” that is indicated.

Line 129: Replace “stud” with “study”.

Line 168: replace “lens.” With “lens”

Statistical analysis:

Replace “double masked” with “double blind”

Line 256-257: The authors should please explain why T test was used for the analysis. Lumping together the various data at the different time point and comparing with baseline does not explain the full picture. “Repeated measures ANOVA” should be appropriate rather. If the authors wanted to only use T Test, then there was no need to take the measures in between the baseline and the 6 months.

Results

Table 1 is very confusing. In the statistical analysis section, the authors have stated this “All continuous variables are expressed as mean ± standard deviation; categorical variables are expressed as absolute numbers and percentages”. However, in Table 1, the authors have presented something different because, the means and standard deviations are not presented. For the age, the authors have presented the mean with no standard deviations. For gender, the authors have presented the actual numbers. For the rest of the parameters, they have presented values I can’t tell if it’s “Mean” or something else. The authors should work on the Table 1.

Line 293-296: The authors should kindly indicate in magnitude the change in MPOD recorded for each eyes. Sometimes, it may not be “statistically significant” but could be “biologically significant”.

Figure 1a and b. The authors have not clearly labelled the figures. Same for Fig. 2a and b and 3a and b and fig. 4. The figures appear blurred.

Discussion

Line 460: The authors stated that The Veggie Meter uses “reflectometry technique”. The Veggie Meter rather uses “reflection spectroscopy”. The authors should kindly correct it.

It is important the authors have mentioned the impact of BMI as a determinant of carotenoid status. My suggestion for future research is to adjust the dosage of the supplement intake to match participants’ weight.

The authors should also discuss the potential impact of “cholecystectomy” on the bioavailability of the carotenoids.

Conclusion

The authors should kindly indicate by what magnitude the intake of the supplements had on the outcomes of interest. For example, they could say MPOD

References

The authors should kindly check reference number 2 and cite its well.

Reviewer 2 Report

In this manuscript, the authors reported that night vision was improved through 6-month carotenoids supplementation in healthy subjects.

The manuscript is properly structured and clearly written. The introduction contains background information and properly leads to the aim of the study. The methods and the statistical tests are clearly described. Although the sample size is small, statistically significant improvements were observed in MPOD and visual functions in the supplemented group. The discussion is consistent with the results. Limitations of the study were also acknowledged.

I have only minor comments:

  1. Introduction
    • Line 42-43 This sentence is confusing. It seems the United States of America is also one of the developing countries. The authors should better rephrase the sentence to improve clarity.
  1. Materials and Methods
    • Line 146 Indentation needed.
  1. Statistical analysis
    • Line 245 Format consistency needed. Suggestion: “3.0 Statistical analysis” should be “3. Statistical analysis”.
    • Line 267-273 The active components in the placebo capsules is unclear. Do the placebo capsules also contain additional antioxidants and minerals as the active capsules? Also, I recommend moving the “Active and Placebo Capsules” section forward for reading easily and not put it under the “3.0 Statistical analysis” section.
  1. Result
    • Line 275-276 Regarding the significant difference in age between the groups, would this affect the results?
    • Table 1 I recommend keeping Table 1 on one page to make it easier to read.
    • Figure 5 Low Luminance Questionnaire. Based on the publication of Owsley et al., “Subscale scores were computed by scaling individual items from 0 to 100, where 100 represents the highest functional level and 0 the lowest, and then averaging the individual items.” Isn’t decrease in LLQ score corresponding to less functionality? Please clarify how do you interpret the y-axis “Low Luminance Questionnaire Total”.
    • Line 385 Format consistency needed. Suggestion: “[Figure 6A]” to “(Figure 6a)”
  1. Appendix 2
    • Line 561 Typo. The word “exxplaining” should be corrected to “explaining”.

Round 2

Reviewer 1 Report

Thank you for revising the manuscript. It is greatly improved now. That notwithstanding, kindly find below some minor comments for your perusal.

Abstract

Line 28: Replace “imporvement” with “improvement”.

Results

The authors have removed Figure 5 which was in the earlier version of the manuscript. They must thus make the current Figure 6 the present Figure 5.

Author Response

We have made the typographical corrections requested by the reviewer and are grateful for the positive comments.

Regards

Pinakin Davey